# Gradient-based Optimization for Compact and Explainable Fuzzy Rule-based Classification

Javier Fumanal-Idocin*[1], Raquel Fernandez-Peralta[2], and Javier Andreu-Perez[1]

[1]School of Computer Science and Electronic Engineering, University of Essex
[2]Mathematical Institute, Slovak Academy of Sciences, Bratislava, Slovakia
 j.fumanal-idocin@essex.ac.uk, raquel.fernandez@mat.savba.sk, j.andreu-perez@essex.ac.uk

## Abstract

Rule-based models are valued in high-stakes decision-making for their transparency, but their discrete nature limits optimization and scalability. We propose the Fuzzy Rule-based Reasoner (FRR), a gradient-based rule learner that enforces user-defined complexity constraints while maintaining strong performance. FRR combines interpretable fuzzy logic partitions with sufficient (single-rule) decision-making, avoiding the combinatorial growth of additive ensembles. Across 40 datasets, FRR outperforms traditional rule-based methods (by about 5% over RIPPER), matches the accuracy of tree-based models like CART with rule bases 90% smaller, and achieves 96% of the accuracy of additive rule-based models while using only 3% of their rule base size.

## 1 Introduction

Deep neural networks excel in handling large volumes of unstructured data, such as images and videos [1], but their lack of transparency limits their use in high-stakes domains like medicine and finance [2]. Rule-based algorithms, by contrast, are inherently interpretable, as they reveal explicit decision patterns and their relevance, offering more trustworthy explanations than many post-hoc XAI methods [3, 4]. Studying learned rules allows practitioners to validate or challenge model insights and even use them to approximate complex models like deep networks [5, 6]. However, rule-based classifiers often face a trade-off between interpretability and accuracy: larger rule sets improve performance but reduce transparency [7]. While fuzzy logic and genetic optimization have been used to balance this trade-off [8], they struggle to scale, and gradient-based approaches [9–13] often produce overly complex rule bases. To address these issues, we propose the Fuzzy Rule-Based Reasoner (FRR), a fully differentiable rule-based classifier that integrates user-defined complexity constraints (maximum rules and conditions per rule) with interpretable fuzzy partitions, achieving a balance between performance, simplicity, and transparency.

---

*Corresponding Author.

## 2 Fuzzy Rule-Based Reasoner

The FRR is a hierarchical model composed of four layers of matrix operations that mimic fuzzy logical reasoning (Figure 1). It receives an input vector $X_i$ and produces a class prediction through a sequence of fuzzification, inference, and decision steps.

1. **Fuzzification layer**: The fuzzification layer transforms input features into interpretable fuzzy membership values. Continuous variables are mapped to linguistic terms such as "low," "medium," and "high" using trapezoidal fuzzy sets defined according to the data's quantile distribution, while categorical variables are encoded using a one-hot representation.

2. **Logic inference layers**: The logic inference process in the FRR has two main steps. First, for each feature, the model selects the most relevant fuzzy label, the one with the highest weight, representing which linguistic partition is activated. Next, it determines which features form the rule's antecedent, keeping a fixed number of conditions per rule. The truth value of a rule is then computed as the product of the selected condition contributions, ensuring the result remains within the $[0, 1]$ range.

3. **Decision layer**: Implements sufficient-rule prediction by selecting the consequent of the highest-scoring activated rule.

Beyond these layers, the FRR also incorporates a parsimony mechanism that automatically prunes irrelevant rule conditions during training. Through a competitive cancellation process, conditions that contribute little to the prediction are deactivated and penalized via a regularization term, ensuring compact and efficient rule bases without sacrificing accuracy.

## 3 Training

Training the FRR addresses three main challenges: non-differentiability, gradient sparsity, and vanishing gradients. The non-differentiable arg max used for rule and feature selection is approximated with the

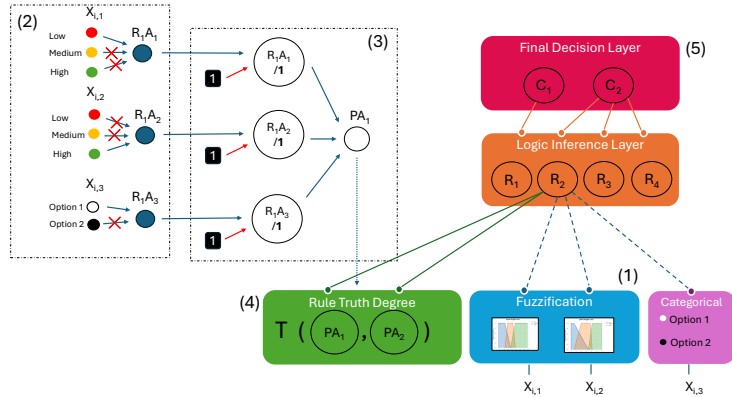

**Figure 1.** FRR scheme, using an input $X_i$ with three features, four rules, and two target classes.

**Table 1.** 5-fold results for all the datasets considered.

| Method | Sufficient Rule-based | | | | | Tree-based | | | Additive Rule-based | | LR | GB |
| --- | --- | --- | --- | --- | --- | --- | --- | --- | --- | --- | --- | --- |
| | FRR | FGA | DRNet | DINA | RIPPER | CART | C4.5 | GOT | SIRUS | RRL | | |
| Accuracy | 79.51 | 70.46 | 56.08 | 54.99 | 75.22 | 81.06 | 79.99 | 76.91 | 82.17 | 81.99 | 82.12 | 86.04 |
| Number of Rules | 13.77 | 7.12 | 24.04 | 18.48 | 16.04 | 39.75 | 131.92 | 5.23 | 286.71 | 99.35 | – | – |
| Conditions/Rule | 1.94 | 2.23 | 6.37 | 4.59 | 1.96 | 5.75 | 8.10 | 2.27 | 2.90 | 8.85 | – | – |
| Rule base Size | 26.71 | 15.87 | 153.13 | 84.82 | 31.43 | 228.56 | 1068.55 | 11.87 | 831.45 | 879.24 | – | – |
| Unique Conditions | 10.78 | 10.52 | 16.26 | 9.18 | 21.30 | 34.72 | 68.56 | 11.00 | 357.05 | 125.16 | – | – |

Straight-Through Estimator (STE) [14, 15], enabling gradient flow through discrete choices. To reduce gradient sparsity, a relaxed indicator function with parameter $\beta$ allows partial updates to non-selected features; $\beta$ gradually decreases during training to recover discrete behavior. Finally, to combat vanishing gradients from multiplicative rule inference, two strategies are applied: root-normalized activation, which rescales small truth values, and residual connections [16], which preserve gradient flow early in training and fade out over time. Together, these mechanisms ensure stable and efficient optimization of interpretable fuzzy rules.

## 4 Experiments

We evaluated the proposed Fuzzy Rule-based Reasoner (FRR) across 40 widely used classification datasets, ranging from 80 to 19,020 samples and 2 to 85 features, using 5-fold cross-validation and accuracy as the primary metric. Statistical differences between classifiers were assessed using the Friedman Test and Post-hoc Nemenyi procedure [17]. The FRR was compared against three groups of baselines: (i) rule-based methods, including the Fuzzy Genetic Algorithm classifier (FGA) [18], RIPPER [19], SIRUS [20], DRNet [9], DINA [21], and Rule-based Representation Learning (RRL) [22]; (ii) tree-based models, such as CART [23], C4.5 [24], and the Generalized Optimal Sparse Decision Tree (GOT) [25]; and (iii) non-rule-based classifiers, namely Logistic Regression (LR) and Gradient Boosting (GB) [26]. While GB achieved the highest average accuracy (86.04%), FRR maintained competitive performance (79.51%) with significantly lower complexity. Indeed, its rule base was only 3% the size of SIRUS and 11% that of CART. FRR was statistically superior to other sufficient rule-based classifiers, particularly RIPPER, and demonstrated scalability across datasets of varying dimensionality. Gradient-based rule learners (DRNet and DINA) performed worse, likely due to their sensitivity to hyperparameters and smaller dataset sizes. Overall, FRR achieved a strong balance between interpretability and accuracy, offering an efficient and explainable alternative to traditional and gradient-based rule learning methods.

## 5 Conclusion

We introduced the Fuzzy Rule-based Reasoner (FRR), an explainable classifier that learns interpretable rules through gradient-based optimization while allowing users to control model complexity by setting limits on rule count and length. This design maintains interpretability without sacrificing performance, achieving a strong balance between accuracy and simplicity. FRR outperforms other gradient-based rule learners and reaches accuracy comparable to tree-based models with far lower complexity. Future work will integrate FRR into deep learning frameworks to provide rule-based explanations within gradient flows and explore how its learned rules relate to epistemic and aleatoric uncertainty.

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
