# OpenReview forum: "Gradient-based Optimization for Compact and Explainable Fuzzy Rule-based Classification"
_NLDL.org/2026/Abstracts_Track — NLDL 2026 Abstracts_

### Official Review · Reviewer_D7Du · 2025-10-24

**Soundness:** 3
**Correctness:** 3
**Rating:** 4
**Confidence:** 4

**Summary:**

The paper introduces a Fuzzy Rule-Based Reasoner (FRR), to account for the lack of explainability in classical ML models such as tree-based methods and gradient-boosting. They introduce a differentiable rule based model that allows gradient learning for interpretable rules. They compare against standard ML methods (as mentioned above) and achieve comparative performance for fewer rules, thereby aiding interpretability.

**Strengths:**

The method seems theoretically sound, particularly in its use of modern mechanisms for handling vanishing/exploding gradients, along with the use of the straight-through estimator.

They compare against similar models and achieve comparative performance with substantially fewer rules applied. They compare across a wide range of datasets, I assume reporting average accuracy across those.

The main contribution is the substantially fewer rules required to achieve comparable performance, and the number of conditions per-rule further underlines this.

**Weaknesses:**

The main weakness lies in the reporting of performance. When reporting accuracy across multiple datasets and folds in a single number i would expect some measure of the variance (CIs) to be included. Additionally, accuracy may not be ideal across that many datasets due to likely label distribution skewness. One could fear that the model with its few rules makes the simple choices of always picking the most frequent classes, thereby achieving an accuracy that is not indicative of general performance.

---

### Official Review · Reviewer_ptYe · 2025-11-03

**Soundness:** 3
**Correctness:** 3
**Rating:** 4
**Confidence:** 3

**Summary:**

This work presents the Fuzzy Rule-based Reasoner (FRR), a gradient-based, interpretable classifier designed to bridge the gap between performance and explainability in rule-based learning. The FRR framework employs differentiable fuzzy logic layers, user defined complexity constraints, and a pruning mechanism to maintain compact, human understandable rule bases. The model addresses key challenges in non-differentiability and gradient sparsity using Straight-Through Estimators, relaxed indicators, and residual connections. Across 40 benchmark datasets, FRR achieves competitive accuracy.

**Strengths:**

- Architecture and learning mechanisms are clearly explained and well-motivated.
- Comprehensive evaluation across several datasets with testing.
- Addresses real world needs for explainable models in high stakes decision domains.

**Weaknesses:**

- Runtime and computational cost of training not quantified and whether this method is practical for large-scale problems.
- While accuracy is covered in depth, the interpretability aspect could be evaluated more rigorously.

---

### Decision · Program_Chairs · 2025-11-05

**Decision:**

Accept

**Comment:**

The abstract is of interest to the community and should be presented at the conference.